# Recursive Inversion Models for Permutations

**Christopher Meek**
Microsoft Research
Redmond, Washington 98052
meek@microsoft.com

**Marina Meilă**
University of Washington
Seattle, Washington 98195
mmp@stat.washington.edu

## Abstract

We develop a new exponential family probabilistic model for permutations that can capture hierarchical structure and that has the Mallows and generalized Mallows models as subclasses. We describe how to do parameter estimation and propose an approach to structure search for this class of models. We provide experimental evidence that this added flexibility both improves predictive performance and enables a deeper understanding of collections of permutations.

## 1 Introduction

Among the many probabilistic models over permutations, models based on penalizing inversions with respect to a reference permutation have proved particularly elegant, intuitive, and useful. Typically these generative models "construct" a permutation in stages by inserting one item at each stage. An example of such models are the *Generalized Mallows Models (GMMs)* of Fligner and Verducci (1986). In this paper, we propose a superclass of the GMM, which we call the *recursive inversion model* (RIM), which allows more flexibility than the original GMM, while preserving its elegant and useful properties of compact parametrization, tractable normalization constant, and interpretability of parameters. Essentially, while the GMM constructs a permutation sequentially by a stochastic *insertion sort* process, the RIM constructs one by a stochastic *merge sort*. In this sense, the RIM is a compactly parametrized *Riffle Independence (RI)* model (Huang & Guestrin, 2012) defined in terms of inversions rather than independence.

## 2 Recursive Inversion Models

We are interested in probabilistic models of permutations of a set of elements $E = \{e_1, ..., e_n\}$. We use $\pi \in \mathbb{S}_E$ to denote a permutation (a total ordering) of the elements in $E$, and use $e_i <_\pi e_j$ to denote that two elements are ordered. We define an $n \times n$ (lower diagonal) *discrepancy matrix* $D_{ij}$ that captures the discrepancies between two permutations.

$$D_{ij}(\pi, \pi_0) = \begin{cases} 1 & i <_\pi j \wedge j <_{\pi_0} i \\ 0 & otherwise \end{cases} \tag{1}$$

We call the first argument of $D_{ij}(\cdot, \cdot)$ the *test permutation* (typically $\pi$) and the second argument the *reference permutation* (typically $\pi_0$).

Two classic models for permutations are the Mallows and the generalized Mallows models. The *Mallows model* is defined in terms of the *inversion distance* $d(\pi, \pi_0) = \sum_{ij} D_{ij}(\pi, \pi_0)$ which is the total number of *inversions* between $\pi$ and $\pi_0$ (Mallows, 1957). The Mallows models is then $P(\pi|\pi_0, \theta) = \frac{1}{Z(\theta)} \exp(-\theta d(\pi, \pi_0))$, $\theta \in \mathbb{R}$. Note that the normalization constant does not depend on $\pi_0$ but only on the *concentration parameter* $\theta$. The *Generalized Mallows model* (GMM) of Fligner and Verducci (1986) extends the Mallows model by introducing a parameter for each of the elements in $E$ and decomposes the inversion distance into a per element dis-

tance[1]. In particular, we define $v_j(\pi, \pi_0)$ to be the number of inversions for element $j$ in $\pi$ with respect to $\pi_0$ is $v_j(\pi, \pi_0) = \sum_{i >_{\pi_0} j} D_{ij}(\pi, \pi_0)$. In this case, the GMM is defined as $P(\pi|\pi_0, \theta) = \frac{1}{Z(\theta)} \exp(-\sum_{e \in E} \theta_e v_e)$ $\theta \in \mathbb{R}^n$. The GMM can be thought of as a stagewise model in which each of the elements in $E$ are inserted according to the reference permutation $\pi_0$ into a list where the parameter $\theta_e$ controls how likely the insertion of element $e$ will yield an inversion with respect to the reference permutation. For both of these models the normalization constant can be computed in closed form

Our RIMs generalize the GMM by replacing the sequence of single element insertions with a sequence of recursive merges of subsequences where the relative order within the subsequences is preserved. For example, the sequence $[a, b, c, d, e]$ can be obtained by merging the two subsequences $[a, b, c]$ with $[d, e]$ with zero inversions and the sequence $[a, d, b, e, c]$ can be obtained from these subsequences with 3 inversions. The RIM generates a permutation recursively by merging subsequences defined by a binary recursive decomposition of the elements in $E$ and where the number of inversions is controlled by a separate parameter associated with each merge operation.

More formally, a RIM $\tau(\theta)$ for a set of elements $E = \{e_1, \ldots, e_n\}$, has a structure $\tau$ that represents a recursive decomposition of the set $E$ and a set of parameters $\theta \in \mathbb{R}^{n-1}$. We represent a RIM as a binary tree with $n = |E|$ leaves, each associated with a distinct element of $E$. We denote the set of internal vertices of the binary tree by $\mathcal{I}$ and each internal vertex is represented as a triple $i = (\theta_i, i_L, i_R)$ where $i_L$ ($i_R$) is the left (right) subtree, and $\theta_i$ controls the number of inversions when merging the subsequences generated from each of the subtrees. Traversing the tree $\tau$ *in preorder*, with the left child preceding the right child induces a permutation on $E$ called the *reference permutation* of the RIM which we denote as $\pi_\tau$.

The RIM is defined in terms of the *vertex discrepancy*, the *number of inversions at (internal) vertex* $i = (\theta_i, i_L, i_R)$ of $\tau(\theta)$ for test permutation $\pi$ is $v_i(\pi, \pi_\tau) = \sum_{l \in L_i} \sum_{r \in R_i} D_{lr}(\pi, \pi_\tau)$ where $L_i$ ($R_i$) is the subset of elements $E$ that appear as leaves of $i_L$ ($i_R$), the left (right) subtree of internal vertex $i$. Note that the sum of the vertex discrepancies over the internal vertices is the inversion distance between $\pi$ and the reference permutation $\pi_\tau$. Finally, the likelihood of a permutation $\pi$ with respect to RIM $\tau(\theta)$ is as follows:

$$P(\pi|\tau) \propto \prod_{i \in \mathcal{I}} \exp(-\theta_i v_i(\pi, \pi_\tau)) \tag{2}$$

**Example:** For elements $E = \{a, b, c, d\}$, Figure 1 shows a RIM $\tau$ for preferences over four types of fruit. The reference permutation for this model is $\pi_\tau = (a, b, c, d)$ and the modal permutation is $(c, d, a, b)$ due to the sign of the root vertex. For test permutation $\pi = (d, a, b, c)$, we have that $v_{root}(\pi, \pi_\tau) = 2$, $v_{left} = 0$, and $v_{right} = 1$. Note that the model captures strong preferences between the pairs $(a, b)$ and $(c, d)$ and weak preferences between $(c, a), (d, a), (c, b)$ and $(d, b)$. This is an example of a set of preferences that cannot be captured in a GMM as choosing a strong preference between the pairs $(a, b)$ and $(c, d)$ induces a strong preference between either $(a, d)$ or $(c, b)$ which differs in both strength and order from the example.

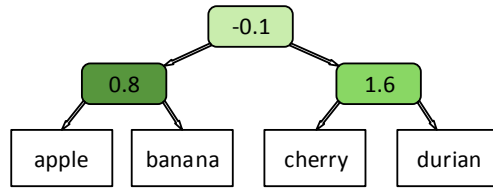

Figure 1: An example of a RIM for fruit preferences among (a)pple, (b)anana, (c)herry, and (d)urian. The parameter for internal vertices indicates the preference between items in the left and right subtree with 0 indicating no preference and a negative number indicating the right items are more preferable than the left items.

Naive computation of the partition function $Z(\tau(\theta))$ for a *recursive inversion model* would require a sum with $n!$ summands (all permutations). We can, however, use the recursive structure of $\tau(\theta)$ to compute it as follows:

**Proposition 1**

$$Z(\tau(\theta)) = \prod_{i \in \mathcal{I}} G(|L_i|, |R_i|; \exp(-\theta_i)). \tag{3}$$

$$G(n, m; q) = \frac{(q)_{n+m}}{(q)_n (q)_m} \overset{def}{\equiv} Z_{n,m}(q). \tag{4}$$

In the above $G(n, m; q)$ is the Gaussian polynomial (Andrews, 1985) and $(q)_n = \prod_{i=1}^{n}(1 - q^i)$. The Gaussian polynomial is not defined for $q = 1$ so we extend the definition so that $G(n, m, 1) = \binom{n+m}{m}$ which corresponds to the limit of the Gaussian polynomial as $q$ approaches 1 (and $\theta$ approaches 0).

Note that when all $\theta_i \geq 0$ the reference permutation $\pi_\tau$ is also a *modal permutation* and that this modal permutation is unique when all $\theta_i > 0$. Also note that a GMM can be represented by using a chain-like tree structure in which each element in the reference permutation is split from the remaining elements one at a time.

## 3 Estimating Recursive Inversion Models

In this section, we present a Maximum Likelihood (ML) approach to parameter and structure estimation from an observed data $\mathcal{D} = \{\pi_1, \pi_2, \ldots \pi_N\}$ of permutations over $E$.

Parameter estimation is straight-forward. Given a structure $\tau$, we see from (2) that the likelihood factors according to the structure. In particular, a RIM is a product of exponential family models, one for each internal node $i \in \mathcal{I}$. Consequently, the (negative) log-likelihood given $\mathcal{D}$ decomposes into a sum

$$- \ln P(\mathcal{D}|\tau(\theta)) = \sum_{i \in \mathcal{I}} \underbrace{[\theta_i \bar{V}_i + \ln Z_{|L_i|, |R_i|}(e^{-\theta_i})]}_{score(i, \theta_i)} \tag{5}$$

with $\bar{V}_i = \frac{1}{|\mathcal{D}|} \sum_{\pi \in \mathcal{D}} v_i(\pi, \pi_\tau)$ representing the sufficient statistic for node $i$ from data. This is a convex function of the parameters $\theta_i$, and hence the ML estimate can be obtained numerically solving a set of univariate minimization problems. In the remainder of the paper we use $D$ to be the sum of the discrepancy matrices for all of the observed data $\mathcal{D}$ with respect to the identity permutation. Note that this matrix provides a basis for efficiently computing the sufficient statistics of any RIM.

In the remainder of this section, we consider the problem of estimating the structure of a RIM from observed data beginning with a brief exploration of the degree to which the structure of a RIM can be identified.

### 3.1 Identifiability

First, we consider whether the structure of a RIM can be identified from data. From the previous section, we know that the parameters are identifiable given the structure. However, the structure of a RIM can only be identified under suitable assumptions.

The first type of non-identifiability occurs when some $\theta_i$ parameters are zero. In this case, the permutation $\pi_\tau$ is not identifiable, because switching the left and right child of node $i$ with $\theta_i = 0$ will not change the distribution represented by the RIM. In fact, as shown by the next proposition, the left and right children can be swapped without changing the distribution if the sign of the parameter is changed.

**Proposition 2** *Let $\tau(\theta)$ be a RIM over $E$, $D$ a matrix of sufficient statistics and $i$ any internal node of $\tau$, with parameter $\theta_i$ and $i_L, i_R$ its left and right children. Denote by $\tau'(\theta')$ the RIM obtained from $\tau(\theta)$ by switching $i_L, i_R$, and setting $\theta'_i = -\theta_i$. Then, $P(\pi|\tau(\theta)) = P(\pi|\tau'(\theta'))$ for all permutations $\pi$ of $E$.*

This proposition demonstrates that the structure of a RIM cannot be identified in general and that there is an equivalence class of alternative structures among which we cannot distinguish. We elimi-

nate this particular type of non-identifiability by considering RIM that are in *canonical form*. Proposition 2 provides a way to put any $\tau(\theta)$ in canonical form.

---

**Algorithm 1** Algorithm CANONICALPERMUTATION

---

**Input** any $\tau(\theta)$
**for** each internal node $i$ with parameter $\theta_i$ **do**
   **if** $\theta_i < 0$ **then**
      $\theta_i \leftarrow -\theta_i$; switch left child with right child
   **end if**
**end for**

---

**Proposition 3** *For any matrix of sufficient statistics $D$, and any RIM $\tau(\theta)$, Algorithm* CANONICALPERMUTATION *does not change the log-likelihood.*

The proof of correctness follows from repeated application of Proposition 2. Moreover, if $\theta_i \neq 0$ before applying CANONICALPERMUTATION, then the output of the algorithm will have all $\theta_i > 0$.

A further non identifiability arises when parameters of the generating model are equal. It is easy to see that if all the parameters $\theta_i$ are equal to the same value $\theta$, then the likelihood of a permutation $\pi$ would be $P(\pi|\tau,(\theta,\dots\theta)) \propto \exp(-\theta d(\pi, \pi_\tau))$, which is the likelihood corresponding to the Mallows model. In this case $\pi_\tau$ is identifiable, but the internal structure is not. Similarly, if all the parameters $\theta_i$ are equal in a subtree of $\tau$, then the structure in that subtree is not identifiable.

We say that a RIM $\tau(\theta)$ is *locally identifiable* iff $\theta_i \neq 0$, $i \in \mathcal{I}$ and $|\theta_i| \neq |\theta_{i'}|$ whenever $i$ is a child of $i'$. We say that a RIM $\tau(\theta)$ is *identifiable* if there is a unique canonical RIM that represents the same distribution. The following proposition captures the degree to which one can identify the structure of a RIM.

**Proposition 4** *A RIM $\tau(\theta)$ is identifiable iff it is locally identifiable.*

### 3.2 ML estimation or $\tau$ for fixed $\pi_\tau$ is tractable

We first consider ML estimation when we fix $\pi_\tau$, the reference permutation over the leaves in $E$. For the remainder of this section we assume that the optimal value of $\hat{\theta}_i$ for any internal node $i$ is available (e.g., via the convex optimization problem described in the previous section). Hence, what remains to be estimated is the internal tree structure

**Proposition 5** *For any set $E$, permutation $\pi_\tau$ over $E$, and observed data $\mathcal{D}$, the Maximum Likelihood RIM structure inducing this $\pi_\tau$ can be computed in polynomial time by Dynamic Programming algorithm* STRUCTBYDP.

**Proof sketch** Note that there is a one-to-one correspondence between tree structures representing alternative binary recursive partitioning over a fixed permutation of $E$ and alternative ways in which the one can parenthesize the permutation of $E$. The negative log-likelihood decomposes according to the structure of the model with the *cost* of a subtree rooted at $i$ depending only on the structure of this subtree. Furthermore, this cost can be decomposed recursively into a sum of $score(i, \hat{\theta}_i)$ and the costs of $i_L, i_R$ the subtrees of $i$. The recursion is identical to the recursion of the "optimal matrix chain multiplication" problem, or to the "inside" part of the Inside-Outside algorithm in string parsing by SCFGs (Earley, 1970).

Without loss of generality, we consider that $\pi_\tau$ is the identity, $\pi_\tau = (e_1, \dots e_n)$. For any subsequence $e_j, \dots e_m$ of length $l = m - j + 1$, we define the variables $cost(j, m), \theta(j, m), Z(j, m)$ that will store respectively the negative log-likelihood, the parameter at the root, and the $Z$ for the root node of the *optimal tree* over the subsequence $e_j, \dots e_m$. If all the values of $cost(j, m)$ are known for $m - j + 1 < l$, then the values of $cost(j, j + l - 1), \theta(j, j + l - 1), Z(j, j + l - 1)$ are obtained recursively from the existing values. We also maintain the pointers $back(j, m)$ that indicate which subtrees were used in obtaining $cost(j, m)$. When $cost(1, n)$ and the corresponding $\theta$ and $Z$ are obtained, the optimal structure and its parameters have been found, and they can be read

recursively by following the pointers $back(j, m)$. Note that in the innermost loop, the quantities $score(j, m), \theta(j, m), \bar{V}$ are recalculated for each $k$.

We call the algorithm implementing this optimization STRUCTBYDP.

---

**Algorithm 2** Algorithm STRUCTBYDP

---

1: **Input** sample discrepancy matrix $D$ computed from the observed data
2: **for** $m = 1 : n$ **do**
3:    $cost(m, m) \leftarrow 0$
4: **end for**
5: **for** $l \leftarrow 2 \ldots n$ **do**
6:    **for** $j \leftarrow 1 : n - l + 1$ **do**
7:       $m \leftarrow j + l - 1$
8:       $cost(j, m) \leftarrow \infty$
9:       **for** $k \leftarrow j : m - 1$ **do**
10:          calculate $\bar{V} = \sum_{j'=j}^{k} \sum_{m'=k}^{m} D_{m'j'}$
11:          $L = k - j + 1, R = m - k$
12:          estimate $\theta_{jm}$ from $L, R, \bar{V}$
13:          calculate $score(j, m)$ by (5)
14:          $s \leftarrow cost(j, k) + cost(k + 1, m) + score(j, m)$
15:          **if** $s < cost(j, m)$ **then**
16:             $cost(j, m) \leftarrow s, back(j, m) \leftarrow k$
17:             store $\theta(j, m), Z_{LR}(j, m)$
18:          **end if**
19:       **end for**
20:    **end for**
21: **end for**

---

---

**Algorithm 3** Algorithm SASEARCH

---

**Input** set $E$, discrepancy matrix $D$ computed from observed data, inverse temperature $\beta$
**Initialize** Estimate GMM $\tau_0$ by BRANCH&BOUND , $\tau^{best} = \tau_0$
**for** $t = 1, 2, \ldots t_{max}$ **do**
  **while** *accept*= FALSE **do**
    sample $\pi \sim P(\pi | \tau_{t-1})$
    $\tau' \leftarrow$ STRUCTBYDP$(\pi, D)$
    $\tau' \leftarrow$ CANONICALPERMUTATION$(\tau')$
    $\pi' \leftarrow$ reference order of $\tau'$
    $\tau' \leftarrow$ STRUCTBYDP$(\pi', D)$
    *accept*=TRUE, $u \sim uniform[0, 1)$
    **if** $e^{-\beta(\ln P(D|\tau_{t-1}) - \ln P(D|\tau'))} < u$ **then**
      *accept*$\leftarrow$ FALSE
    **end if**
  **end while**
  $\tau_t \leftarrow \tau'$ (store accepted new model)
  **if** $P(D|\tau_t) > P(D|\tau^{best})$ **then**
    $\tau^{best} \leftarrow \tau_t$
  **end if**
**end for**
**Output** $\tau^{best}$

---

To evaluate the running time of STRUCTBYDP algorithm, we consider the inner loop over $k$ for a given $l$. This loop computes $\bar{V}, \hat{\theta}, Z$ for each $L, R$ split of $l$, with $L + R = l$. Apparently, this would take time cubic in $l$, since $\bar{V}$ is a summation over $LR$ terms. However, one can notice that in the calculations of *all* $\bar{V}$ values over this submatrix of size $l \times l$, for $L = 1, 2, \ldots l - 1$, each of the $D_{rl}$ elements is added once to the sum, is kept in the sum for a number of steps, then is removed. Therefore, the total number of additions and subtractions is no more than twice $l(l-1)/2$, the number of submatrix elements . Estimating $\theta$ and the score involved computing $Z$ by (3) (for

the score) and its gradient (for the $\theta$ estimation). These take $\min(L, R) < l$ operations per iteration. If we consider the number of iterations to convergence a constant, then the inner loop over $k$ will take $\mathcal{O}(l^2)$ operations. Since there are $n - l$ subsequences of length $l$, it is easy now to see that the running time of the whole STRUCTBYDP algorithm is of the the order $n^4$.

### 3.3 A local search algorithm

Next we develop a local search algorithm for the structure when a reference permutation is not provided. In part, this approach can be motivated by previous work on structure estimation for the Mallows model, where the structure is a permutation. For these problems, researchers have found that an approach in which one greedily improves the log-likelihood by transposing adjacent elements coupled with a good initialization is a very effective approximate optimization method (Schalekamp & van Zuylen, 2009; Ali & Meila, 2011).

In our approach, we take a similar approach and treat the problem as a search for good reference permutations leveraging the STRUCTBYDP algorithm to find the structure given a reference permutation. At a high level, we initialize $\pi_\tau = \pi_0$ by estimating a GMM from the data $D$ and then improve $\pi_\tau$ by "local changes" starting from $\pi_0$.

We rely on estimation of a GMM for initialization but, unfortunately, the ML estimation of a Mallows model, as well as that of a GMM, is NP-hard (Bartholdi et al., 1989). For the initialization, we can use any of the fast heuristic methods of estimating a Mallows model, or a more computationally expensive search algorithm, The latter approach, if the search space is small enough, can find a provably optimal permutation but, in most cases, it will return a suboptimal result.

For the local search, we make two variations with respect to the previous works, and we add a local optimization step specific to the class of Recursive Inversion models. First, we replace the greedy search with a simulated annealing search. Thus, we will generate proposal permutations $\pi'$ near the current $\pi$. Second, the proposals permutations $\pi'$ are not restricted to pairwise transpositions. Instead, we sample a permutation $\pi'$ from the current $RIM$ $\tau_t$. The reason is that if some of the pairs $e \prec_{\pi_\tau} e'$ are only weakly ordered by $\tau_t$ (which would happen if this ordering or $e, e'$ is not well supported by the data), then the sampling process will be likely to create inversions between these pairs. Conversely, if $\tau_t$ puts a very high confidence on $e \prec e'$, then it is probable that this ordering is well supported by the data and reversing it will be improbable in the proposed $\tau$.

For each accepted proposal permutation $\pi$, we estimate the optimal structure $\tau$ give this $\pi$ and the optimal parameters $\hat{\theta}$ given the structure $\tau$. Rather than sampling a permutation from the RIM $\tau(\hat{\theta})$ we then apply CANONICALPERMUTATION, which does not change the log-likelihood, to convert $\tau(\hat{\theta})$ into a canonical model and perform another structure optimization step STRUCTBYDP. This has the chance of once again increasing the log-likelihood, and experimentally we find that it often does increase the log-likelihood significantly. We then use the estimated structure and associated parameters to sample a new permutation. These steps are implemented by algorithm SASEARCH.

## 4 Related work

In addition to the Mallows and GMM models, our RIM model is related to the work of Manilla & Meek (2000). To understand the connection between this work and our RIM model consider a restricted $RIM$ model in which parameter values can either be 0 or $\infty$. Such a model provides a uniform distribution over permutations consistent with a *series-parallel partial order* defined in terms of the binary recursive partition where a parameter whose value is 0 corresponds to a parallel combination and a parameter value of $\infty$ corresponds to a series combination. The work of Manilla & Meek (2000) considers the problem of learning the structures and estimating the parameters of mixtures of these series-parallel RIM models using a local greedy search over recursive partitions of elements.

Another close connection exists between RIM  models and the *riffle independence models (RI)* proposed by Huang et al. (2009); Huang & Guestrin (2012); Huang et al. (2012). Both approaches use a recursive partitioning of the set of elements to define a distribution over permutations. Unlike the RIM model, the RI  model is not defined in terms of inversions but rather in terms of independence between the merging processes. The RI model requires exponentially more parameters than the

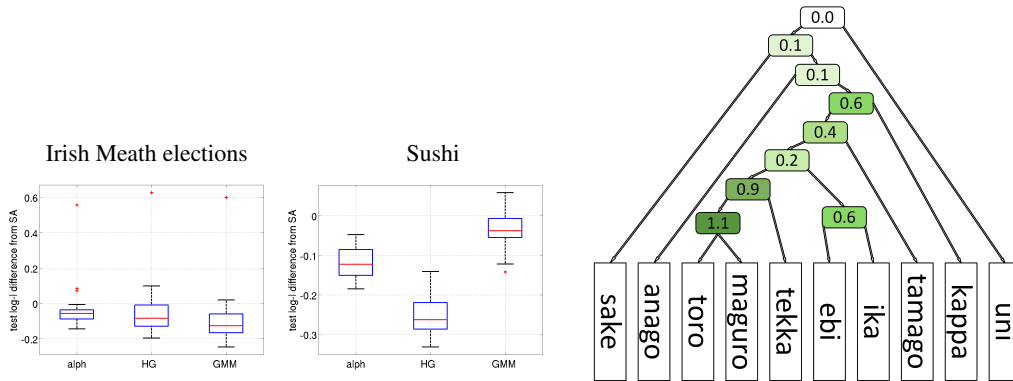

Figure 2: Log-likelihood scores for the models `alph`, `HG`, and `GMM` as differences from the log-likelihood of SASEARCH output, on held-out sets from Meath elections data (left) and Sushi data (middle). Train/test set split was $90/2400$, respectively $300/4700$, with 50 random replications. Negative score indicate that a model has lower likelihood than the model obtained by SASEARCH. The far outlier(s) in `meath` represent one run where SA scored poorly on the test set. Right: Most common structure and typical parameters learned for the sushi data. Interior nodes contain the associated parameter value, with higher values and darker green indicating a stronger ordering between the items in the left and right subtrees. The leaves are the different types of sushi.

RIM model due to the fact that the model defines a general distribution over mergings which grows exponentially in the cardinality of the left and right sets of elements. In addition, the RI models do not have the same ease of interpretation as the RIM model. For instance, one cannot easily extract a reference permutation or modal permutation from a given a RI model, and the comparison of alternative RI models, even when the two RI models have the same structure, is limited to the comparison of rank marginals and Fourier coefficients.

It is worth noting that there have been a wide range of approaches that use multiple reference permutations. One benefit of such approaches is that they enable the model to capture multi-modal distributions over permutations. Examples of such approaches include the mixture modeling approaches of Manilla & Meek (2000) discussed above and the work of Lebanon & Lafferty (2002) and Klementiev et al. (2008), where the model is a weighted product of a set of Mallows models each with their own reference order. It is natural to consider both mixtures and products of RIM models.

## 5   Experiments

We performed experiments on synthetic data and real-world data sets. In our synthetic experiments we found that our approach was typically able to identify both the structure and parameters of the generative model. More specifically, we ran extensive experiments with $n = 16$ and $n = 33$, choosing the model structures to have varying degrees of balance, and choosing the parameters randomly chosen with $exp(-\theta_i)$ between $0.4$ and $0.9$. We then used these RIMs to generate datasets containing varying numbers of permutations to investigate whether the true model could be recovered. We found that all models were recoverable with high probability when using between 200-1000 SASEARCH iterations. We did find that the identification of the correct tree structure in its entirety typically required a large sample size. We note that failures to identify the correct structure were typically due to the fact that alternative structures had higher likelihood than the generating structure in a particular sample rather than a failure of the search algorithm. While our experiments had at most $n = 33$ this was not due to the running time of the algorithms. For instance, STRUCTBYDP ran in a few seconds for domains with 33 items. For the smaller domains and for the real-world data below, the whole search with hundreds of accepted proposals typically ran in less than three minutes. In particular, this search was faster than the BRANCH&BOUND search for GMM models.

In our experiments on real-world data sets we examine two datasets. The first data set is an Irish House of Parliament election dataset from the Meath constituency in Ireland. The parliament uses the single transferable vote election system, in which voters rank candidates. There were 14 candi-

dates in the 2002 election, running for five seats. Candidates are associated with the two major rival political parties, as well as a number of smaller parties. We use the roughly 2500 fully ranked ballots from the election. See Gormley & Murphy (2007) for more details about the dataset. The second dataset consists of 5,000 permutations of 10 different types of sushi where the permutation captures preferences about sushi (Kamisha, 2003). The different types of sushi considered are: anago (sea eel), ebi (shrimp), ika (squid), kappa-maki (cucumber roll), maguro (tuna), sake (salmon), tamago (egg), tekka-maki (tuna roll), toro (fatty tuna), uni (sea urchin).

We compared a set of alternative *recursive inversion models* and approaches for identifying their structure. Our baseline approach, denoted `alph`, is one where the reference permutation is alphabetical and fixed and we estimate the optimal structure given that order by STRUCTBYDP. Our second approach, `GMM`, is to use the BRANCH&BOUND algorithm of Mandhani & Meila (2009)[2] to estimate a generalized Mallows Model. A third approach, `HG`, is to fit the optimal RIM parametrization to the hierarchical tree structure identified by Huang & Guestrin (2012) on the same data.[3] Finally, we search over both structures and orderings with SASEARCH, with 150 (100) iterations for Meath (sushi) at temperature $0.02$.

The quantitative results are shown in Figure 2. We plot the difference in test log-likelihood for each model as compared with SASEARCH. We see that on the Meath data SASEARCH outperforms `alph` in 94% of the runs, `HG` in 75%, and `GMM` in 98%; on the Sushi data, SASEARCH is always superior to `alph` and `GMM`, and has higher likelihood than `GMM` in 75% of runs. On the training sets, SASEARCH had always the best fit (not shown).

We also investigated the structure and parameters of the learned models. For the Meath data we found that there was significant variation in the learned structure across runs. Despite the variation there were a number of substructures common to the learned models. Similar to the findings in Huang & Guestrin (2012) on the structure of a learned riffle independence model, we found that candidates from the same party were typically separated from candidates of other parties as a group. In addition, within these political clusters we found systematic preference orderings among the candidates. Thus, many substructures in our trees were also found in the `HG` tree. In addition, again as found by Huang & Guestrin (2012), we found that a single candidate in an extreme political party is typically split near the top of the hierarchy, with a $\theta \approx 0$, indicating that this candidate can be inserted anywhere in a ranking. We suspect that the inability of a `GMM` to capture such dependencies leads to the poor empirical performance relative to `HG` and full search which can capture such dependencies. We note that `alph` is allowed to have $\theta_i < 0$, and therefore the alphabetic reference permutation does not represent a major handicap.

For the sushi data roughly 90% of the runs had the structure shown in Figure 2 with the other variants being quite similar. The structure found is interesting in a number of different ways. First, the model captures a strong preference between different varieties of tuna (toro, maguro and tekka) which corresponds with the typical price of these varieties. Second, the model captures a preference against tamago and kappa as compared with several other types of sushi and both of these varieties are distinct in that they are not varieties of fish but rather egg and cucumber respectively. Finally, uni (sea urchin), which many people describe as being quite distinct in flavor, is ranked independently of preferences between other sushi and, additionally, there is no consensus on its rank.

## Footnotes

[1]Note that a GMM can be parameterized in terms of $n - 1$ parameters due to the fact that $v_n = 0$.

[2]`www.stat.washington.edu/mmp/intransitive.html`

[3]We would have liked to make a direct comparison with the algorithm of Huang & Guestrin (2012), but the code was not available. Due to this, we aim only at comparing the quality of the `HG` structure, a structure found to model these data well albeit with a different estimation algorithm, with the structures found by SASEARCH.

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
