[Reviews · NeurIPS 2014]

Submitted by Assigned_Reviewer_20

Mallows models are a classically studied class of distributions over
permutations that can be viewed as a sequential model in which items
are inserted one by one into a ranking. This paper proposes an interesting
hierarchical generalization of Mallows models in which groups of items
are sequentially ``merged'' together (as they would be in mergesort).
The model can also be viewed as a special case of a recently proposed
class of ``riffle independent'' models by Huang/Guestrin, but with
a more tractable number of parameters in general and better computational
properties.

There are several nice contributions in this paper, including a simple
and elegant characterization of identifiability of the structure, as well as
an interesting structure estimation algorithm
based on the inside-outside parsing algorithm for stochastic context free
grammars.

The authors also discuss a few results on Sushi and Irish election datasets
which seem promising. I think that the tree visualizations of the distributions
are particularly intuitive, and should be an easy way to do a
a preliminary analysis of any new ranking dataset.
I would have been interested in the synthetic data experiments too, but these did not seem discussed.
Though now there may be a lot of new space for the authors to discuss these (see below).

It is interesting that the authors were able to reproduce the clusters that were found in
the Huang/Guestrin analysis --- which suggests a much simpler form of interaction was
sufficient to capture most of the variation. I would love to see a discussion of
what cannot be captured well by the proposed RIM model.

Overall, the paper is well written, highly technical, and treats
an interesting class of models that should have broad application
in ranking. Given the structure of the model, this paper should also
lead to a number of easy extensions including mixtures models and
learning with partially ranked data (in the style of the Lu/Boutilier,
Huang et al, or Lebanon/Mao papers).
I heartily recommend acceptance to NIPS.

some small errors:
grep for: paramter, matricies, ``If if'', appraoch, interpretabtion

The algorithm for SASEARCH seems to appear twice in the paper.
The authors could get significantly more space by deleting one :)
Summary: The paper proposes a nice new class of models for permutations that generalizes the classic Mallows model. These models provide a very intuitive way to understand ranking datasets.

Submitted by Assigned_Reviewer_43

This paper introduces a new probability model (and prior) for permutations. The model is hierarchical (a binary tree) and can thus capture certain "group" preferences between the elements. The authors do a good job introducing prior work on Mallows models and putting their new model into context. The description of their propositions and algorithms is well structured and easy to follow. The concept seems sound and is interesting.
However, the experimental section is comparably weak. Experiments where only performed on two datasets and no direct comparison to prior work and established models is presented. The results mostly show how different choices for the reference permutation \pi_r affect the results and thus how different variants of the proposed algorithm compare to each other.

I also find it very hard to judge the significance of this work: While the concept is interesting, it is not obvious that the proposed methods result in better or more interpretable models.

Figure 2: I don't understand how the level 3 and level 7 nodes of the tree can have 1 and 3 children repectively. Don't the described algorithms operate on binary trees only?

Typos:
- Page 2, line 56: D_{i,j}(...)
- Page 8, line 390: identifying the *their* structure
Summary: An Ok paper but the experimental section is rather weak and does not present any direct comparisons to current state of the art.

Submitted by Assigned_Reviewer_44

The authors claim that they extend the Generalized Mallow Models (GMMs) by considering hierarchical structure formulated in equation (2). An example is provided in Figure 1 to show the motivation. The proposed formulation eq(2) is very similar to an existing work Extended Mallows Models (EMM) (see eq(5) in [2*] and eq(10) in [1*]). Considering structure for Mallow Models may bring benefits, and it would be good if compared with [1*]. Also the writing can be improved.

Missing references:
[1*] G. Lebanon and J. Laferty, Cranking: combining ranking using conditional probability models on permutations, icml 2002.
[2*] A. Klementiev, D. Roth, K. Small, Unsupervised rank aggression with distance-based models, icml 2008.
Summary: The paper has a reasonable idea, and its writing can be improved.
Author Feedback
Author rebuttal: Thanks to all reviewers for the time they took to evaluate our paper and for
the feedback that will help improve this paper!

Reviewer 44:

The authors are pleased to discuss these papers ([1*]Lebanon & Lafferty -
Cranking, [2*] Klementiev & al -- Extended Maloows Model (EMM)) now that they
will have more space :)

The similarity between our work and the above papers is only via their comon ancestors, the papers of Fligner and Verducci and possibly Critchlow (cited in the EMM paper). Here are the main differences:

- both [1*] and [2*] use the Kendall distance, where all inversions are weigthed by the same parameter theta. This is a true metric. [1*,2*] extend the Mallows model by adding new permutations sigma_i with one corresponding theta_i each.
We, on the other hand, stay with the single sigma (the reference permutation), but augment the number of theta parameters to n-1, so that individual groups of inversions are penalized differently.

- our model is graphical, in the sense that there is a dependency
structure and a reference permutation that can be read directly from the tree structure (when tau is canonical, identifyable). The RIM is unimodal (if identifyable). In contrast [1*,2*] are specifically *designed* to allow multimodality, and there is no general way to relate the dependency structure to the sigma_i's. This just shows that RIM and the EMM/Cranking address different types of distributions over permutations. == > Hence, on data, one would expect RIM to perform well when EMM does not, and viceversa.

On the other hand, the distance between partial rankings introduced in [2*] con be seamlessly adapted with the RIM model, if one wanted to use it for partial rankings. We will describe this possible extension in the final paper.

Reviewer 43:

Figure 2: this is a graphical typo -embarassing!- maguro should link to the
1.1 node, "anago" should link to the lower 0.1 node.

"No comparison to prior work or established models is presented." == This
statement is false.==

The most related models are the GMM (very established) and the RI
model (a superclass, very well studied in several papers). The direct
comparison with GMM are presented in Figure 2. As for the RI model,
footnote 4 explains that we did the closest possible indirect
comparison.

"..how different choices of the reference permutation pi_r affect the results..."
The reviewer is mistaken: our algorithm ESTIMATES pi_r; one thing we
show is that if we don't, the results are worse. It's not the most
important conclusion to draw from the experiments, but there it is.

In addition, by chosing the same dependency structure as HG for our
data, we "show" that: SA can also find this structure or one very
similar, without independence tests. Show is in "" because with the
richer parametrization of the RI model it's never completely sure that
there could not be other independencies modeled but not captured by
the tree.

The other existing models for permutations (see also response to reviewer 44) do not penalize inversions differentially, but penalize all inversions between a pi and a sigma by the same parameter theta (or 0 in the case of some models like Lebanon and Mao). Therefore they would be at a disadvantage in the case of the sushi data for example. However, we could easily compare with some of these models when we write an extended version of the paper.

We have other real data experiments that were not shown (e.g jester
with 70 items) because of lack of space.

Reviewer 20:

About the experiments on artificial data: We found the model is recoverable w.h.p (in 200-1000 SA iterations) if identifyable and the sample size sufficiently large.

The most interesting empirical observation is that balanced trees and extremely unbalanced trees (corresponding to the GMM) seem not make much difference for structure estimation. I.e neither cathegory is much harder than the other. So, for a tree with 32 nodes, where the first split is 16+16, the algorithm may
occasionally make mistakes, but no more mistakes than in the structure
of a GMM over 32 nodes, where each split has a "thin" branch. This is not what happens with the algorithm of Huang et al, that exploits "thin" branches.

About what the RIM model won't capture: the RIM model has (i) certain
obvious independencies, and also (ii) obvious parameter sharing (which
induces dependences on the probability of inversion between some pairs
of variables). Here are some examples referring to Fig 1. Because of
(i), the ordering of c,d will be independent of the ordering of a,b
regardless of what the data says. Because of (ii), P[a < c]
(a before c) and P[b < d] cannot be freely chosen, as knowing one, plus the
other parameters (0.8 and 1.6) will determine the other. In particular
for Fig 1, they have to be both near 0.5